# Mechanical Properties and Antibacterial Effect on Mono-Strain of *Streptococcus mutans* of Orthodontic Cements Reinforced with Chlorhexidine-Modified Nanotubes

**DOI:** 10.3390/nano12172891

**Published:** 2022-08-23

**Authors:** Elias Nahum Salmerón-Valdés, Ana Cecilia Cruz-Mondragón, Víctor Hugo Toral-Rizo, Leticia Verónica Jiménez-Rojas, Rodrigo Correa-Prado, Edith Lara-Carrillo, Adriana Alejandra Morales-Valenzuela, Rogelio José Scougall-Vilchis, Alejandra Itzel López-Flores, Lia Hoz-Rodriguez, Ulises Velásquez-Enríquez

**Affiliations:** 1Center for Research and Advanced Studies in Dentistry, Faculty of Dentistry, School of Dentistry, Autonomous University of Mexico State, Toluca 50130, Mexico; 2Infectious Diseases Research Unit of the Mexico Children’s Hospital Federico Gómez, Mexico City 06720, Mexico; 3Center for Applied Physics and Advanced Technology, National Autonomous University of Mexico, A.P. 1-1010, Queretaro 76000, Mexico; 4Periodontal Biology Laboratory, School of Dentistry, National Autonomous University of Mexico, Mexico City 04510, Mexico

**Keywords:** glass ionomer cements, chlorhexidine, nanotubes, microhardness, compressive strength

## Abstract

Recently, several studies have introduced nanotechnology into the area of dental materials with the aim of improving their properties. The objective of this study is to determine the antibacterial and mechanical properties of type I glass ionomers reinforced with halloysite nanotubes modified with 2% chlorhexidine at concentrations of 5% and 10% relative to the total weight of the powder used to construct each sample. Regarding antibacterial effect, 200 samples were established and distributed into four experimental groups and six control groups (4 +ve and 2 −ve), with 20 samples each. The mechanical properties were evaluated in 270 samples, assessing microhardness (30 samples), compressive strength (120 samples), and setting time (120 samples). The groups were characterized by scanning electron microscopy and Fourier transform infrared spectroscopy, and the antibacterial activity of the ionomers was evaluated on *Streptococcus mutans* for 24 h. The control and positive control groups showed no antibacterial effect, while the experimental group with 5% concentration showed a zone of growth inhibition between 11.35 mm and 11.45 mm, and the group with 10% concentration showed a zone of growth inhibition between 12.50 mm and 13.20 mm. Statistical differences were observed between the experimental groups with 5% and 10% nanotubes. Regarding the mechanical properties, microhardness, and setting time, no statistical difference was found when compared with control groups, while compressive strength showed higher significant values, with ionomers modified with 10% concentration of nanotubes resulting in better compressive strength values. The incorporation of nanotubes at concentrations of 5% and 10% effectively inhibited the presence of *S. mutans*, particularly when the dose–response relationship was taken into account, with the advantage of maintaining and improving their mechanical properties.

## 1. Introduction

Conventional glass ionomer cements (GICs) are a restorative material and the first choice of cement in dentistry. Specifically, type I ionomers are optimal for the adhesion of orthodontic restorations or bands used in conventional orthopedic treatments. Due to their excellent properties including biocompatibility, a desirable thermal expansion coefficient, and good adherence to enamel and dentin, type I ionomers provide extraordinary clinical benefits. However, the accumulation of dentobacterial plaque around orthodontic bands, along with microfiltration, facilitates the passage of oral fluids and bacteria to the dental tissue, frequently generating lesions by demineralization or white spot lesions, caries, and periodontal disorders, causing a risk for patients receiving orthodontic treatment. Recently, hybrid materials have been developed to improve their properties, including resin-modified glass ionomers (RMGIs), which have been analyzed in various studies and are mainly characterized by anticariogenic activity, an ability to remineralize dentin, and resistance to fracture [1,2,3,4,5,6,7]. However, to date, it has not been possible to increase the antibacterial capacity of these materials, to reduce the number of lesions by demineralization or the recurrence of caries [6,7,8,9,10,11].

Research has been conducted to evaluate the antibacterial capacity of type I glass ionomers with respect to cariogenic microorganisms [8]. Bacteria that have shown greater proliferation in patients with fixed appliances, mainly orthodontic bands, are principally *Streptococcus mutans (S. mutans)* and *Porphyromonas gingivalis*. Specifically, *S. mutans* is the microorganism most frequently involved in the development of carious lesions [3,7].

Recently, nanotechnology has been introduced in the area of dental materials with the aim of improving various properties of these materials. Various antimicrobial components, such as triclosan, fluoride, chlorhexidine (CX), and xylitol, have been incorporated into different dental materials to improve their antibacterial activity [12,13,14]. However, previous studies have been reported that direct incorporation of these antimicrobial components altered the mechanical properties of dental materials [15]. Halloysite nanotubes (HNs) are clay nanostructures with high levels of mechanical strength, thermal stability, and biocompatibility. Their main advantage over other nanocarriers is their low cost. Due to their internal tubular structure, they can be loaded with different drugs for slow release through nanopores located at their ends, prolonging the time of action [13,16,17,18,19]. Some studies have indicated that drugs released from HNs can last 30–100 times longer than the drug alone [17].

HNs can improve the beneficial properties of dental materials [14,20], without altering their mechanical properties. The present study performed different tests to characterize and evaluate the antibacterial effect on *Streptococcus mutans*, and the mechanical properties (microhardness, compressive strength, and setting time) of conventional and hybrid type I glass ionomers modified with and without HNs loaded with CX. The first hypothesis of the study was that the incorporation of HNs with CX into glass ionomer would confer antibacterial effects on these materials. The second hypothesis was that a higher quantity of HNs incorporated into glass ionomers would increase the antibacterial effects conferred on these materials. The third hypothesis was that the incorporation of HNs into ionomers would not negatively affect their mechanical properties. The null hypothesis would be accepted if an absence of an antibacterial effect were observed in experimental groups, or if the dose-response of modified HNs in glass ionomers showed no changes in antibacterial effect, or if the mechanical properties of experimental groups were altered negatively compared with the control groups.

## 2. Materials and Methods

### 2.1. Modification of Halloysite Nanotubes with Chlorhexidine

One gram of HNs (Sigma–Aldrich, St. Louis, MO, USA) that had been previously dried in a HERAtherm drying oven (Thermo Fisher Scientific, Waltham, MA, USA) was weighed using an analytical balance (Shimadzu Scientific Instruments, Kyoto, Japan). A solution of 3-(trimethoxysilyl) propyl-methacrylate-98% (Sigma–Aldrich, St. Louis, MO, USA) diluted to 5% and 95% acetone (Sigma–Aldrich, St. Louis, MO, USA) was used for the immersion of the nanotubes for 24 h at 110 °C in a drying oven.

Subsequently, 1 g of silanized nanotubes was mixed with 10 mL of 2% CX (Consepsis, Ultradent Products, South Jordan, UT, USA), commonly used for disinfection in dentistry [21], and 10 mL of 95% pure ethanol and sonicated for 1 h. The CX-loaded nanotubes were then placed in a drying oven for 10 days at 30 °C to eliminate residual solvent [17].

### 2.2. Incorporation of Modified Nanotubes into Glass Ionomers

In this study, a conventional glass ionomer KC (Ketac Cem, 3M ESPE, Minnesota, USA) and a resin-modified glass ionomer FO (Fuji Ortho, GC CORPORATION, Tokyo, Japan) were used. Two hundred blocks, with a diameter of 3 mm and thickness of 1 mm, were fabricated in a Teflon matrix. The materials were handled according to the manufacturer’s instructions. The materials were activated with light and polymerized by an LED device (Elipar, 3M ESPE, Saint Paul, MN, USA) for 40 s; a Demetron LED radiometer (Kavo Kerr, Charlotte, USA) was used to verify that the minimum intensity of light emitted was 400 mW/cm^2^.

The amount of powder recommended by the manufacturer was weighed into 10 samples, using the spoon provided by the manufacturer, to determine the average weight. KC showed an average of 0.3631 g, while FO showed an average of 0.2543 g. Once these averages were obtained, 5% and 10% of the powder in each sample was replaced with HNs with and without loaded CX, to form the experimental and positive control groups, respectively. In previous studies, nanostructures including nanotubes have been incorporated to dental materials at percentages from 3 to 20% [12,14,22,23] Some studies have mentioned that a concentration of 10% is necessary to improve the mechanical properties [24]. The modified powder was mixed following the manufacturer’s instructions.

Forty KC and FO ionomer blocks were used for the control group. As a positive control, four groups of ionomers were formed with HNs without loaded CX (80 blocks), which were distributed as follows: KC5HN (KC with 5% HNs), KC10HN (KC with 10% HNs), FO5HN (FO with 5% HNs), and FO10HN (FO with 10% HNs).

For the experimental group, 80 ionomer blocks with HNs loaded with CX were used, which were distributed in the following groups: KC5CX (KC with 5% HNs with CX), KC10CX (KC with 10% HNs with CX), FO5CX (FO with 5% HNs with CX), and FO10CX (FO with 10% HNs with CX). The distribution can be observed in Figure 1. A total of 200 circumferential blocks (5 mm *×* 1 mm) were fabricated to evaluate antibacterial effect.

### 2.3. Sample Characterization with Fourier Transform Infrared Spectroscopy (FTIR)

Fourier transform infrared (FTIR) analysis or FTIR spectroscopy was employed to determine the presence of chlorhexidine in experimental groups, and to compare their chemical properties with the control group. The samples were analyzed on a 6700 FTIR spectrometer (Perkin Elmer, Waltham, MA, USA) by ATR (attenuated total reflectance) using a diamond/ZnSe crystal plate. Thirty-two scans were performed on each sample at spectral resolution of 5 cm^−1^ with an infrared spectrum range of 400 to 4000 cm^−1^.

### 2.4. Scanning Electron Microscopy (SEM)

The specimens were mounted and observed by a cold field emission scanning electron microscope (Hitachi SU8230, Hitachi High-Technologies Corporation, Tokyo, Japan) at 1.0 keV equipped with a Bruker XFlash 6/60.

### 2.5. Microbiology Assay

The microbiological tests performed in this study were carried out according to the guidelines established in standard M100 of the Clinical and Laboratory Standards Institute (CLSI). *Streptococcus mutans (S. mutans)* 33688 (ATCC) was seeded in Petri dishes with Muller Hinton agar (MHA) supplemented with 5% sheep blood (BD Columbia II, Germany) using the cross-streaking technique, and incubated at 37 °C for 18 h. Five colonies were taken from the fresh culture and adjusted to the 0.5 turbidity standard of the McFarland nephelometer (1.5 × 10^8^ CFU/mL) with 0.9% NaCl_2_ solution for dilutions [25].

Subsequently, the MHA Petri dishes were inoculated with *S. mutans,* and the glass ionomer blocks corresponding to each group were added. The plates were placed in an incubator (Thermo Fisher Scientific, Massachusetts, USA) for 24 h at 37 °C in an anaerobic atmosphere with 5% CO_2._ The entire procedure was performed in triplicate.

After the plates were removed from the incubator, they were examined to verify that the bacterial growth was uniform. the bacterial inhibition was evaluated by measuring the zones of growth inhibition in millimeters with a Vernier caliper, taking into account the diameter of the glass ionomer blocks.

### 2.6. Microhardness

The Vickers scale based on ISO 9917-1: 2007 standards was used to evaluate microhardness. The sample was conformed of 5 circumferential blocks (10 mm *×* 3 mm) for each Ketac and Fuji group (KC, KC5CX, KC10CX, FO, FO5CX and FO10CX). A total of 30 blocks with 25 indentations per block were used to evaluate microhardness. The blocks were placed on the microdurometer (SXHV-1000TA, Sinowen, Dongguan, China) and a force of 10 Newtons for 10 s was applied, using a diamond indenter certified by ISO 9001:2008.

### 2.7. Compression Strength

For evaluation of compressive strength, 120 rectangular blocks (4 mm *×* 3 mm *×* 3 mm) were fabricated, divided into six groups previously mentioned with 20 samples in each. The blocks were analyzed with a universal testing machine (Autograph AGS-X, Shimadzu Corporation, Tokyo, Japan); the flat tip was placed in the center of the sample, and the formula CS = 2P/πdh was used to calculate the compressive strength, where CS represents the compressive strength, P is the load at the fracture, d is the width of the sample and h is the thickness of the sample. The results were obtained in MPa based on the ISO 9917:1991 standard.

### 2.8. Setting Time

Experimental and control pastes were placed into 120 rectangular molds (4 mm *×* 3 mm *×* 3 mm) divided as previously described. The setting time was measured according the ISO method for water-based dental cement (ISO 9917-1:2007) recording the time elapsed between the start of mixing and the moment where the needle (1.06 mm diameter and 400 g weight established in the indenter) did not mark the surface with a complete circular indentation.

The mechanical properties data obtained from the microbiological assay were analyzed using the statistical program IBM SPSS statistical software (Version 25, IBM Corporation, New York, NY, USA). Shapiro–Wilk, Kruskal–Wallis and the Mann–Whitney U test were performed to evaluate the inhibitory effect of the experimental and control groups. For the evaluation of microhardness, setting time, and compressive strength in the experimental and control groups, Shapiro–Wilk, one way ANOVA, and Tukey testing were used.

## 3. Results

### 3.1. IR Spectroscopy

The IR spectra (Figure 2) of halloysite showed absorption bands at 746 cm^−^^1^ corresponding to hydroxyl groups (OH) and at 908 cm^−^^1^ corresponding to Al–OH stretching; the vibrational bands at 1006 cm^−^^1^ and 1119 cm^−^^1^ were attributed to silicate groups (Si–O–Si and Si–O_2_, respectively), and two bands at 3621 cm^−^^1^ and 3697 cm^−^^1^ were related to Si–O–Al groups.

The presence of CX was corroborated by the N_2_–C=N stretching band at approximately 1642 cm^−^^1^, and vibrational bands at 1348 cm^−^^1^ and 2866 cm^−^^1^ related to CH_2_ and CH methyl groups, respectively (Figure 2).

In the FO sample spectra (Figure 3), the vibrational bands at 1725 cm^−^^1^ and 1584 cm^−^^1^ were attributed to C=O polymeric acid carboxyl groups. The peak at 1538 cm^−^^1^ was related to the stretching of C=C double bonds, reflecting the increased interaction between the glass ionomer and the HNs at a concentration of 10%. The peaks at 2979 and 2883 cm^−^^1^ were attributed to CH_2_ and CH_3_ methyl groups, and silicon group peaks were observed at 1068 and 998 cm^−^^1^, corresponding to Al–O–Si and Si–O–Si, respectively, corroborating the presence of HNs in the FO glass ionomer at concentrations of 5% and 10%.

Regarding the KC spectra (Figure 4), bands associated with aluminum polyacrylate C=O were found at 1570 cm^−^^1^ and 1454 cm^−^^1^, the peak at 1395 cm^−^^1^ was attributed to methyl CH, and the peaks at 2887 cm^−^^1^ and 2981 cm^−^^1^ were found to correspond to CH_2_ and CH_3_, respectively. The SiO_2_ group was clearly observed in the peaks at 1156 cm^−^^1^ and 1026 cm^−^^1^, which may correspond to the increase in HNs in the KC group at 5%.

### 3.2. SEM Results

The HN micrograph shows symmetrical, agglomerated, and disorganized nanotubes with an average size of 200–500 nm in length and a width of approximately 50 nm (Figure 5).

The image of the FO control group (Figure 6a) shows irregular areas with a rough surface, in addition to particles where the surface is smoother, with an approximate dimension of 5 to 10 µm.

In the micrograph of the FO group with HNs (Figure 6b), irregular, agglomerated particles are observed, as well as small spherical particles of approximately 1–2 µm.

The photomicrograph corresponding to the KC group (Figure 7a) shows an irregular surface with particles ranging from 5 to 10 µm.

In the image of the KC group with HNs (Figure 7b), agglomerated particles with an approximate size of 1 to 2 µm are visible.

### 3.3. Microbiology Assay

The control and positive control groups did not show inhibition of *S. mutans*. However, the four experimental groups showed an antibacterial effect on this microorganism, with a mean between 11.35 mm and 13.2 mm.

Figure 8 shows that the FO10CX group had a mean of 12.45 mm, one more millimeter of inhibitory effect than the FO5CX group, which had a mean of 11.45 mm. The difference between the KC5CX and FO5CX groups was only 0.10 mm, which indicated no significant change between the two ionomers. However, the KC10CX group had the greatest inhibitory effect, with a mean of 13.20 mm, a difference of 1.85 mm compared with the KC5CX group, which could be due to the higher percentage of incorporated nanotubes. After 72 h, bacterial growth was observed in all experimental groups.

The normality of the data was verified with the Shapiro–Wilk test. Regarding the inhibitory effect of the experimental and control groups, a Kruskal–Wallis test was performed for multiple comparisons, and the Mann–Whitney U test was used to analyze differences between two groups. On the other hand, microhardness, setting time, and compressive strength of the experimental and control groups showed a normal distribution; they were compared with one way ANOVA testing and Tukey testing to analyze differences between groups.

Statistically significant differences were observed between the experimental groups evaluated in this study, using the Kruskal–Wallis test with a value of *p* = 0.001 (Table 1).

A pairwise comparison was performed with the Mann–Whitney U test to determine the differences between the four experimental groups. Statistically significant differences were observed in all groups that contained a higher percentage of nanotubes (10%) compared with those that contained 5%, which suggests that greater inhibitory effect is obtained when the percentage of nanotubes is increased (Table 2). The greatest differences were between the FO5CX and KC10CX groups, and between the KC5CX and KC10CX groups, yielding *p* = 0.001. Finally, when the KC10CX group was compared with the FO10CX group, a minimal difference of 0.75 mm was observed, which indicates no significant difference between the groups loaded at 10% (*p* = 0.223).

Descriptive results from the mechanical tests and setting times for control and experimental groups analyzed in this study can be observed in Table 3. In terms of microhardness, statistically significant differences were not observed when the Fuji control (FO) was compared with experimental groups FO5CX and FO10CX, with means of 68.83, 67.96, and 67.66 respectively, *p =* 0.766. In the same way, Ketac Cem control (KC) was compared with experimental groups KC5CX and KC10CX and no statistically significant differences were observed from the ANOVA test (*p* = 0.056), with means of 80.03, 77.87, and 77.66 respectively. The results for setting time were equal at 7.56 min for the Ketac Cem groups, and 9.57 min. for FO and FO5CX, while only FO10CX was different with 9.56 min. However, statistically significant differences were not observed.

On the other hand, values of compressive strength were observed to increase in all experimental groups (FO5CX, FO10CX, KC5CX, KC10CX), and statistically significant differences were indicated by ANOVA testing with a *p* value of 0.001 (Table 3). In Table 4, a Tukey test comparison shows statistically significative differences between all the study groups.

## 4. Discussion

Certain treatments require the use of orthodontic bands due to the stability these provide to appliances, despite the disadvantages of hindering oral hygiene, causing the accumulation of dentobacterial plaque and giving rise to white lesions [26,27].

An investigation by Tasios et al. [28] mentioned that 24% of teeth treated with orthodontics developed at least one white spot, with the maxillary and mandibular first molars being most affected. Therefore, different alternatives have been implemented with the objective of reducing the presence of these lesions. Among the main alternatives are methods to improve the bactericidal properties of dental materials. However, an appropriate material has not yet been found that can act as a bactericide in the mouth and is efficient as a cementing agent for orthodontic bands [29,30].

Previous studies have concluded that glass ionomers show bacterial inhibition due to the release of fluoride [31]. Several studies have mentioned that this antibacterial effect occurs at a minimum concentration of 5000 parts per million (ppm) [32]. Other studies report that glass ionomers alone are capable of releasing fluoride between 32.6 and 17.4 ppm [33]. Therefore, despite being efficient as restoratives, bases, or cementing agents, these dental materials are limited in their antibacterial effect. Several studies have developed different materials and compounds (nanoparticles of hidroxiapatite, fluorapatite and TiO_2_, fiberglass, zirconia, amino acids, chloroxylenol, boric acid and thymol, triclosan, silver nanoparticles, etc.) with the objective of enhancing the mechanical and antibacterial properties of glass ionomer cements. However, those studies indicated that adding some secondary filler into the glass ionomer cements improved some properties and altered others [34]. In the present investigation, an enhancement of antibacterial and mechanical properties was obtained by using CX preloaded on HNs.

Recently, HNs have been incorporated into different dental materials with the aim of improving the materials’ properties. Previous studies concluded that HNs are excellent nanocarriers for drugs, as well as fillings for restorations, because they very efficiently promote the physical and chemical effects of dental materials. However, these nanostructures do not present bacterial inhibition by themselves [18]. In the present investigation, the control groups for glass ionomers without HN loading corroborated the null capacity of bacterial inhibition.

Degrazia et al. [17] incorporated triclosan-loaded HNs into dental resins, and the results demonstrated their efficacy and potential antibacterial effects. In the present investigation, HNs were loaded with CX and subsequently incorporated into type I glass ionomers, and the antibacterial effects of these dental materials were evident in all experimental groups.

An inhibitory effect was observed in all the experimental groups analyzed in this study, showing zones of growth inhibition between 11 mm and 13 mm with loads of 5% and 10% CX respectively, which agrees with previous studies in which CX treatment of *Streptococcus mutans* resulted in slightly lower zones of growth inhibition, between 7 mm and 9 mm [35]. *Streptococcus mutans* (ATCC 33688) was solely used in the present study, due to the ability of the organism to inhabit and invade various areas of the oral cavity making it a prime perpetrator of tooth decay. Moreover, scientific investigators have validated the dominance of *S. mutans* on the depressions of the tooth surface, constituting 39% of streptococci in the oral ecosystem, and their production of glucan is directly proportional to the extent of biofilm formation [36].

Previous studies have stated that is necessary to chose an antimicrobial agent for addition to a restorative material that will provide effective antibacterial action without adversely affecting the material’s mechanical properties [37]. Takahashi conducted a study in which CX was directly incorporated into a glass ionomer, and the results obtained were similar to those of the present work: notably, a greater antibacterial effect was observed when the CX concentration increased. In the present study, the antibacterial effect of ionomers with higher concentrations of nanotubes preloaded with CX increased the zone of growth inhibition. It is important to mention that in the study conducted by Takahashi, increasing the CX concentration affected physical properties [15]. According to our results, most of the mechanical properties analyzed in this study were not negatively affected and compressive strength (CS) was positively modified.

Microhardness (VHN) values of control groups analyzed in this study were similar to values observed in previous studies where conventional and resin-modified glass ionomer cements have been evaluated [38,39,40], and the microhardness of experimental groups was not found significantly altered. The setting time (ST) was consistent with that reported by the manufacturer in control and experimental groups, although some studies mention that incorporation of nanostructures (magnesium nanoparticles) to glass ionomer cements increased the ST in this material [37].

Regarding CS, previous studies found similar values to those observed in conventional and RMGI cements analyzed in our study [41,42,43,44]. On the other hand, in this study the experimental groups with 5% and 10% of Halloysite nanotubes preloaded with chlorhexidine showed an increase in CS, which can be considered a positive modification. These results are similar to reported in previous studies where mechanical properties were improved by adding different nanostructures [23]. Specifically, CS of glass ionomers was improved with the addition of magnesium oxide nanoparticles, although an increase in the setting time was observed, probably because the presence of magnesium ions may impede or interfere with the acid-base reaction [37]. Other studies added 5%, 10%, 15%, and 20% of ceramic powder to glass ionomer cements to improve the CS of the material, concluding that only at a concentration of 10% was it possible to improve the CS without compromising the ST [24]. In our study, concentrations of 5% and 10% halloysite nanotubes with chlorhexidine added to glass ionomer cements increased the CS without compromising the ST.

In a study by Pazourkova et al. [45] in 2019, the presence of CX was corroborated by IR spectroscopic analysis: methylene CH stretching at 2940 cm^−^^1^ and 2860 cm^−^^1^, N_2_–C=N– stretching at approximately 1646 cm^−^^1^, and CH_2_ groups at 1492 cm^−^^1^ were observed. The spectral results for CX in this study were similar, showing peaks for CH at 2866 cm^−^^1^, N_2_–C=N– at 1642 cm^−^^1^, and CH_2_ stretching at 1348 cm^−^^1^.

In 2018, Zhang et al. [46] confirmed the presence of HNs by observing bands corresponding to the hydroxyl group stretching at 3697 cm^−^^1^ and 3624 cm^−^^1^, and Si–O–Si stretching at approximately 1036 cm^−^^1^, while the bands below 1000 cm^−^^1^ corresponded to the symmetrical stretching of Si–O or Al–O groups. Similar results were observed in the current investigation: –OH groups were identified at 3697 cm^−^^1^ and 3621 cm^−^^1^, Si–O–Si was observed at 1006 cm^−^^1^, and finally, Al–O stretching was observed at 908 cm^−^^1^.

With respect to the analysis of glass ionomers through IR spectroscopy, previous studies reported a peak at 3354 cm^−^^1^, corresponding to the stretching of OH groups in the ionomer liquid, a peak at 1705 cm^−^^1^ corresponding to C=O, and C=C stretching observed at 1640 cm^−^^1^ [47]. These results are similar to those observed in the present study, with the OH peak at 3697 cm^−^^1^, C=O peak at 1725 cm^−^^1^, and C=C peak at 1538 cm^−^^1^; the latter group may be associated with the increased interaction of HNs with the glass ionomer at 10% loading.

Previous studies have reported aromatic CH stretching bands at 2914 and 2852 cm^−^^1^ that correspond to an ionomer. The above coincides with the CH groups observed in the present study, where aromatic CH stretching was observed at 2979 and 2883 cm^−^^1^ [48].

The presence of HNs in the glass ionomers was corroborated by the presence of bands at 1068 and 998 cm^−^^1^, corresponding to aluminum (Al–O–Si) and silica (Si–O–Si) stretching, respectively, in the FO ionomer. For the KC ionomer, stretching was observed at 1073 and 950 cm^−^^1^.

The null hypothesis was rejected because all experimental groups showed antibacterial effect, the higher concentration of nanotubes with chlorhexidine in glass ionomer cements showed an increase in antibacterial effect, whilst microhardness and setting time were not altered, and compressive strength was enhanced. Previous studies investigated the mechanical properties of dental adhesives modified with nanotubes [12,13], and others investigated the mechanical properties of resin-based materials modified with nanotubes [17], however, glass ionomers modified with halloysite nanotubes have not been evaluated, nor their antibacterial effect. Some studies evaluated the antibacterial effect of composites modified with clay nanotubes, without evaluating the mechanical properties (microhardness, compressive strength and setting time) [16,19]. Other studies evaluated antimicrobial activity on different dental materials using only one strain of *Streptococcus mutans* as the most common bacterial strain that causes dental caries [1,6,7,9,16,32]. In this study, the mechanical properties and antibacterial effect on *Streptococcus mutans* were evaluated in glass ionomer cements modified with halloysite nanotubes pre-loaded with chlorhexidine.

### Limitations of the Study

The present study was limited through the evaluation of one bacterial strain (*Streptococcus mutans*). Evaluation of the effects on more bacterial strains would have been more suitable, other oral bacteria should be considered for proper microbiological analysis; nevertheless, the authors considered that *S. mutans* is one of the most significant causative of caries-related pathologies. Mechanical properties were limited to three parameters due to the high cost analysis of each.

## 5. Conclusions

Authors concluded that the addition of nanotubes preloaded with chlorhexidine into glass ionomers at concentrations of 5% and 10% showed a notable inhibitory effect on *S. mutans*, without altering the microhardness and setting time. Furthermore, compressive strength was experimentally enhanced using 10% nanotubes. These results suggest that halloysite nanotubes added to conventional and resin-modified glass ionomer cements could be novel method to counteract injuries caused by orthodontic bands, with the advantage of maintaining and improving mechanical properties.

## Figures and Tables

**Figure 1 nanomaterials-12-02891-f001:**
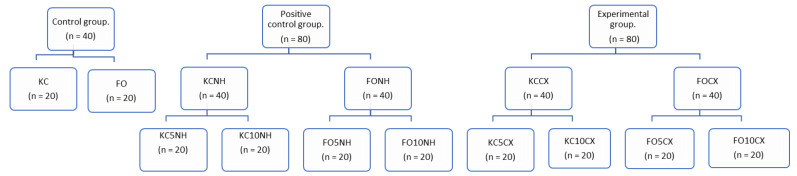
Sample distribution diagram for evaluation of antibacterial effect.

**Figure 2 nanomaterials-12-02891-f002:**
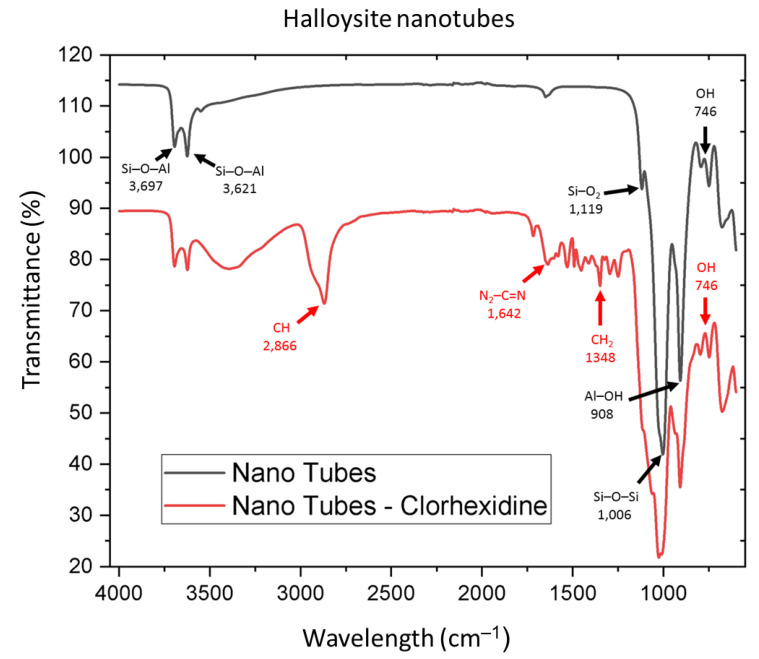
Spectroscopy of halloysite nanotubes and chlorhexidine nanotubes.

**Figure 3 nanomaterials-12-02891-f003:**
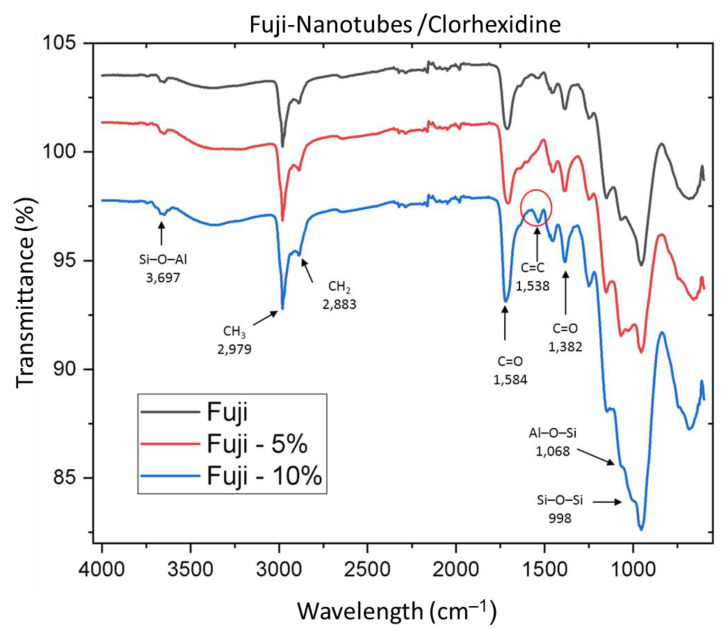
Fuji Ortho (FO) sample spectra.

**Figure 4 nanomaterials-12-02891-f004:**
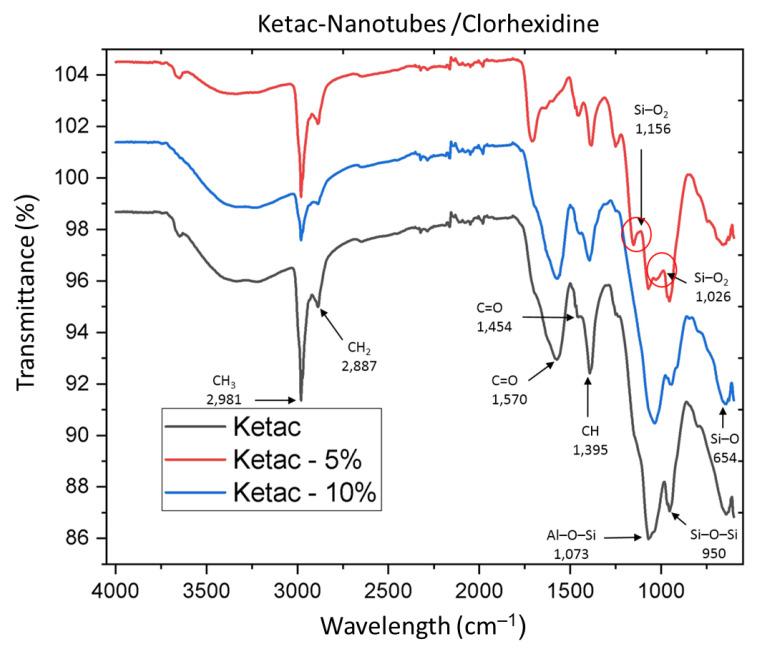
Fuji Ortho sample spectra.

**Figure 5 nanomaterials-12-02891-f005:**
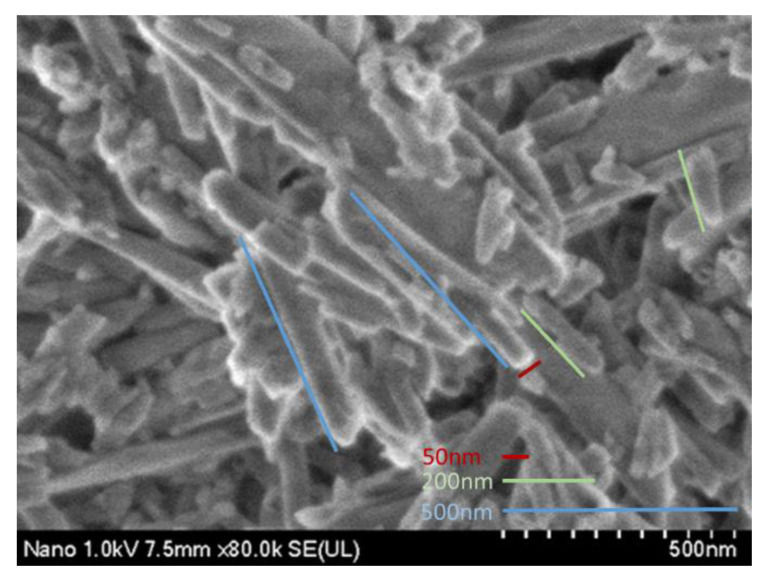
Halloysite nanotubes.

**Figure 6 nanomaterials-12-02891-f006:**
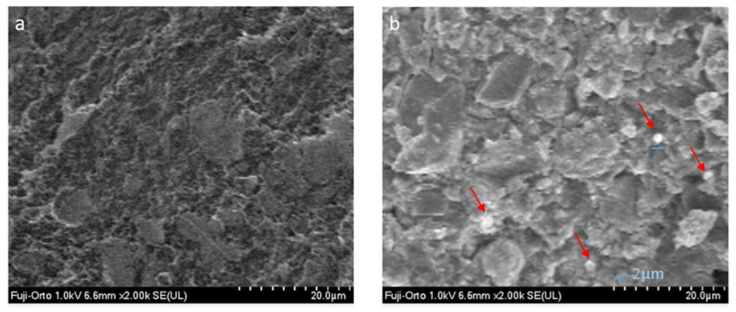
(**a**) Fuji Ortho group, (**b**) Fuji Ortho group with halloysite nanotubes.

**Figure 7 nanomaterials-12-02891-f007:**
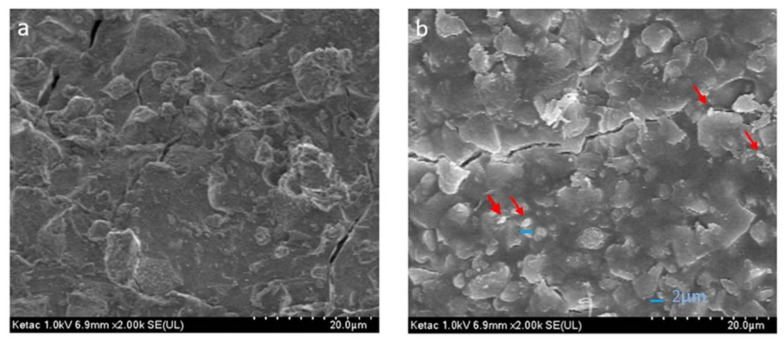
(**a**) Ketac Cem group, (**b**) Ketac Cem group with halloysite nanotubes.

**Figure 8 nanomaterials-12-02891-f008:**
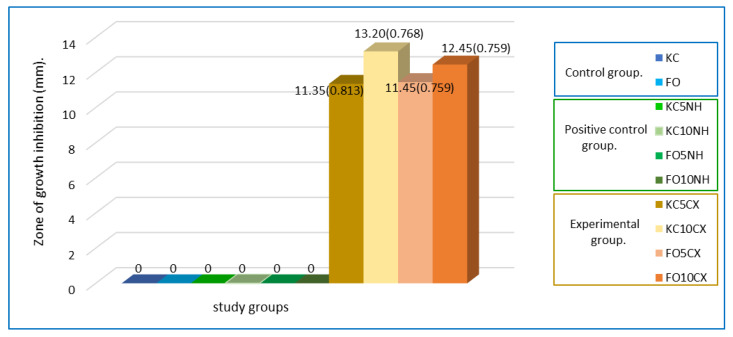
Inhibitory effect of the different groups analyzed in this study; mean (standard deviation); mm: millimeters.

**Table 1 nanomaterials-12-02891-t001:** Comparison of inhibitory effect from experimental groups analyzed in this study.

Groups	Mean (SD)
Fuji ortho 5% of NH + CX. (FO5CX)	11.45 (0.759)
Fuji Ortho 10% of NH + CX. (FO10CX)	12.45 (0.759)
Ketac Cem 5% of NH + CX. (KC5CX)	11.35 (0.813)
Ketac Cem 10% of NH + CX. (KC10CX)	13.20 (0.768)
Total samples	80
Contrast Statistics	41.735
Degrees of freedom	3
*p* value Kruskal Wallis Test	0.001 *

SD: Standard deviation, *: significative differences *p* ≤ 0.05.

**Table 2 nanomaterials-12-02891-t002:** Comparison between experimental groups.

Groups	Contrast Statistics	Contrast Statistics Deviation	*p* Value
KC5CX-FO5CX	−1.725	−0.244	1.000
KC5CX-FO10CX	−24.025	−3.400	0.004 *
KC5CX-KC10CX	−38.760	−5.485	0.001 *
FO5CX-FO10CX	−22.300	−3.156	0.010 *
FO5CX-KC10CX	37.025	5.240	0.001 *
FO10CX-KC10CX	14.725	2.084	0.223

FO5CX: Fuji Ortho 5% of NH + CX; FO10CX: Fuji Ortho 10% of NH + CX; KC5CX: Ketac Cem 5% of NH + CX; KC10CX: Ketac Cem 10% of NH + CX; *: significative differences *p* ≤ 0.05.

**Table 3 nanomaterials-12-02891-t003:** Comparison of mechanical properties from experimental and control groups.

Groups	VMHN Mean (SD)	ST Mean (SD)	CS Mean (SD)
KC (control group)	80.03 (4.56)	7.56 (0.024)	84.16 (0.92)
KC5CX	77.87 (3.63)	7.56 (0.017)	88.78 (1.12)
KC10CX	77.66 (2.99)	7.56 (0.019)	93.96 (1.66)
Total degrees of freedom	74	59	59
Sum of squares	1119.245	0.025	1056.608
Fisher’s statistic	2.992	0.697	293.83
ANOVA test FO groups	0.056	0.502	0.001 *
FO (control group)	68.83 (5.26)	9.57 (0.011)	125.42 (1.79)
FO5CX	67.96 (5.85)	9.57 (0.014)	128.26 (2.26)
FO10CX	67.66 (6.50)	9.56 (0.018)	133.17 (2.13)
Total degrees of freedom	74	59	59
Sum of squares	2520.976	0.014	860.412
Fisher’s statistic	0.267	1.043	71.39
ANOVA test FO groups	0.766	0.359	0.001 *

KC: ketac cem cement, FO: Fuji Orto cement, 5CX: 5% of Chlorhexidine-modified nanotubes, 10CX: 10% of Chlorhexidine-modified nanotubes VMHN: Vickers microhardness, CS: Compressive strength, ST: Setting time, SD: Standard deviation, *: *p* ≤ 0.05 (significative differences).

**Table 4 nanomaterials-12-02891-t004:** Comparison of compressive strength between experimental and control groups.

Groups	Mean Difference	95% Confidence Intervals	*p* Value
Control-KC5CX	−4.62800	IL: −5.6022, SL: −3.6538	0.001 *
Control-KC10CX	−9.80900	IL: −10.7832, SL: −8.8348	0.001 *
KC5CX-KC10CX	−5.18100	IL: −6.1552, SL: −4.2068	0.001 *
Control-FO5CX	−2.84500	IL: −4.4242, SL: −1.2658	0.001 *
Control-FO10CX	−7.75100	IL: −9.3302, SL: −6.1718	0.001 *
FO5CX-FO10CX	−4.90600	IL: −6.4852, SL: −3.3268	0.001 *

* *p* value ≤ 0.05, KC: ketac cem cement, FO: Fuji Orto cement, 5CX: 5% of Chlorhexidine-modified nanotubes, 10CX: 10% of Chlorhexidine-modified nanotubes, inferior limit (IL) and superior limit (SL).

## Data Availability

Not applicable.

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
