# Peer review of "Mechanical Properties and Antibacterial Effect on Mono-Strain of Streptococcus mutans of Orthodontic Cements Reinforced with Chlorhexidine-Modified Nanotubes"

_nanomaterials, 2022, doi:10.3390/nano12172891_

Round 1

Reviewer 1 Report

The authors have satisfactorily addressed the comments of the reviewer.

Author Response

Point 1.

The authors have satisfactorily addressed the comments of the reviewer.

The authors appreciate the reviewer comments.

Reviewer 2 Report

The revised paper  has a more suitable title and abstracts and introdusce mechanical tests with statistical analysis. Generally  seems to be a better manuscript,  but despite  some improvements according to the indicated comments  I do believe that is a need for a second revised manuscript taking into account the following:

„If the research focuses on antimicrobial properties, authors should submit studies comparing several strains of S. mutants”

This comment was indicated in the first revision, but the authors in the revised form  present the same experiments  only with S mutants. They considered as a motivation in the subchapter Limitation of the study „that S. Mutans is one of the most significant causa tive of caries related pathologies, other oral bacteria could be considered for a proper microbiological analysis” In my opinion such motivation is not enough.

The reviewer indicated minor revision, but for a paper with antimicrobial properties I do feel that is mandatory to complete the experimental microbiological part with other strain. 

Other comment for the present form is related to the novelty of the subject which needs as well to be better sustained comparing in detail the existing paper in the literature with the present manuscript

Author Response

Authors would like to thank the Reviewer for his comments

Regarding the following comments:

Point 1: If the research focuses on antimicrobial properties, authors should submit studies comparing several strains of S. mutants”

This comment was indicated in the first revision, but the authors in the revised form present the same experiments only with S mutants. They considered as a motivation in the subchapter Limitation of the study „that S. Mutans is one of the most significant causative of caries related pathologies, other oral bacteria could be considered for a proper microbiological analysis” In my opinion, such motivation is not enough.

The reviewer indicated minor revision, but for a paper with antimicrobial properties, I do feel that is mandatory to complete the experimental microbiological part with other strain.

Other studies have worked with only one strain to evaluate the antibacterial properties of different modified dental materials, described in manuscript lines 568-570. However, authors considered the reviewer comments and recognize the limitation of study to generalize the results like antimicrobial properties; therefore, the title has been modified to “Mechanical properties and antibacterial effect on mono-strain Streptococcus Mutans of orthodontic cements reinforced with chlorhexidine-modified nanotubes.” and in manuscript the content referred to antibacterial properties has been changed to the antibacterial effect on Streptococcus Mutans.

Point 2: Other comment for the present form is related to the novelty of the subject which needs as well to be better sustained comparing in detail the existing paper in the literature with the present manuscript.

The novelty of study was described and compared with previous studies in manuscript lines  561-568.

Best regards

Reviewer 3 Report

The manuscript has been significantly improved and enriched compared to the previous version. New markings have been added. All changes suggested by the reviewer have been implemented.

After reading the manuscript, some minor remarks arise:

Unfortunately, microbiological tests are still performed on 1 strain (several S. mutans strains could be compared).

The chapter on methodology does not describe the statistical analysis.

Figure 8 - Photos are of no use to work.

line 272-274 - this should be transferred to the methodology.

Author Response

Authors would like to thank the Reviewer for his comments

Regarding the following comments:

Point 1: Unfortunately, microbiological tests are still performed on 1 strain (several S. mutans strains could be compared).

Other studies have worked with only one strain to evaluate the antibacterial properties of different modified dental materials, described in manuscript lines 568-570. However, authors considered the reviewer comments and recognize the limitation of study to generalize the results like antimicrobial properties; therefore, the title has been modified to “Mechanical properties and antibacterial effect on mono-strain Streptococcus Mutans of orthodontic cements reinforced with chlorhexidine-modified nanotubes.” and in manuscript the content referred to antibacterial properties has been changed to the antibacterial effect on Streptococcus Mutans.

Point 2: The chapter on methodology does not describe the statistical analysis.

A description of statistical analysis was added to the manuscript on methodology.

Point 3: Figure 8 - Photos are of no use to work.

Figure 8 was eliminated.

Point 4: line 272-274 - this should be transferred to the methodology.

Authors would like to mention that the manuscript lines mentioned by the reviewer correspond to the results obtained through SEM characterization of halloysite nanotubes. Therefore, authors included in results section. However, if you consider that these results must be described in methodology section we are willing to change them.

Best regards.

This manuscript is a resubmission of an earlier submission. The following is a list of the peer review reports and author responses from that submission.

Round 1

Reviewer 1 Report

The present study aims to determine the antibacterial effect of conventional and hybrid type I glass  ionomers modified with Halloysite nanotubes (HN) loaded with chlorhexidine (CX). I have the following comments for the authors to consider:

Introduction

What are the advantages of incorporating CX and HNs in GIC over directly incorporating CX into GIC?

Pls add null hypothesis of the study to Introduction

Materials and methods

Section 2.2 Incorporation of modified nanotubes into glass ionomers

How was the concentration of HN (5% and 10%) determined?

How was the loading of CX into HNs determined?

Section 2.3 Sample characterization with Fourier transform infrared spectroscopy (FTIR)

Pls specify the model/manufacturing company/country for Perkin Elmer spectrophotometer.

ection 2.4 Scanning electron microscopy

Pls specify the model/manufacturing company/country for field emission scanning electron microscope.

Section 2.5 Microbiology assay

Why only mono-species, Streptococcus Mutans being assessed?

Section 2.6 Statistical Analysis

Pls specify the data to be analyzed.

Pls include data on CHX release from GIC over time as it affects the antibacterial properties of GIC.

Results 

Table 1

Pls show the results of statistical analysis in Table 1.

Discussion

Pls include a statement whether the null hypothesis could or could not be rejected.

Pls address the limitations of the study.

What are the advantages of using Halloysite nanotubes as carriers for CX in GIC.

Reviewer 2 Report

The manuscript is worthy of investigation and  suitable for the journal Nanomaterials, with a title conform to the content. The abstract is informative and the paper  is  clearly written and well  organized  with methods able  to sustain the research. A statistical analysis was performed  as well The references are from both parts older and new respectively.  Before publication in my opinion is mandatory to be revised according to the following:

a)       Taking into account that ionomer cements reinforced with chlorhexidine-modified  nanotubes have been investigated recently (2020) in the journal Dental Materials ( reference 12 )it is a need to present in details was is really new in the present paper

b)      The research design will be good to be enriched

c)       the microstructural features are not visible, labeled with arrows and consistent with the paper’s text descriptions of them

d)      in the line 312 ” Some orthopedic treatments” probably needs to be replaced due to the fact that paper is not about such treatments

e)       the conclusion are base on the results existing in the paper but is a need to be rewrite after more results

Reviewer 3 Report

Manuscript "Antibacterial activity and characterization of orthodontic glass ionomer cements reinforced with chlorhexidine-modified nanotubes" presents interesting research results, but requires some additions before publication.

Detailed comments:

In microbiology, zones of growth inhibition are used more often than halo.

line 63-65 - authors should consider providing a more detailed research objective.

If the research focuses on antimicrobial properties, authors should submit studies comparing several strains of S. mutants

What was the diameter of the glass ionomer blocks?

Have the diameter of the glass ionomer blocks been subtracted from the zones of growth inhibition?

line 125 - NaCl or NaCl2 was used for dilutions?

Figure 9 - the results should have the same approximation, there is no marking of standard deviations and no description of the OY axis. Additionally, the statistical analysis should be marked next to the results and not as a separate subchapter.

The Conclusion section should be expanded.